# Numerical Investigation on Effect of Opening Ratio on Structural Performance of Reinforced Concrete Deep Beam Reinforced with CFRP Enhancements

Yasar Ameer Ali [1], Lateef Najeh Assi [2,*], Hussein Abas [3], Hussein R. Taresh [2], Canh N. Dang [4] and SeyedAli Ghahari [5]

1   Department of Construction and Building Engineering Techniques, Al-Mustaqbal University College, Najaf Street, Hilla 51001, Babylon, Iraq
2   Department of Civil Engineering, Mazaya University College, Zaitoon Distract, Nasiriyah 64001, Iraq
3   School of Civil Engineering, Southwest Jiaotong University, Chengdu 610032, China
4   Department for Management of Science and Technology Development, Ton Duc Thang University, Ho Chi Minh City 700000, Vietnam
5   Department of Civil and Environmental Engineering, Purdue University, 600 Purdue Memorial Mall, West Lafayette, IN 47906, USA
*   Correspondence: lateeftiger2000@gmail.com or lateef.asi.ism@atu.edu.iq

**Abstract:** Reinforced concrete deep beams are a vital member of infrastructures such as bridges, shear walls, and foundation pile caps. Thousands of dollars and human lives are seriously threatened due to shear failure, which have developed in deep beams containing web openings. This paper investigates numerically the overall behavior of simply supported concrete deep beams reinforced with carbon fiber-reinforced polymer (CFRP) sheets through forty specimens grouped in four groups. The numerical analysis results agreed well with the experimental results in the literature, particularly the visual failure initiation with a failure load difference of nearly 7%. Finite element analyses indicated that the presence of an opening with considerable width reduced the failure load by about 71% compared to the corresponding solid specimens. In addition, the reinforced concrete deep beam samples started to behave differently when the (b/h) ratio increased more than (2.0). The findings showed that the compression stress strut pathway had been disrupted by the web opening leading to stress redistribution, and the structure will behave as two separate members. Thus, the upper web-opening part sustained the most stress, while the part under the web-opening did not show any stress concentration. The numerical stress distribution results showed that the attributed reason is that rebars and openings helped redirect the stresses to the compression strut. Using CFRP sheets with a width of more than 160 mm significantly improved the reinforced concrete deep beam with web-opening due to the increasing confinement to the upper part of the reinforced concrete deep beams with the opening.

**Keywords:** carbon fiber reinforced polymer (CFRP); strut-and-tie modeling (STM); reinforced concrete deep beam; stress distribution analyses; enhancement of deep beam with openings



## 1. Introduction

Reinforced concrete (RC) deep beams are one of the most common structural components in modern infrastructure construction, typically used when heavy load redistribution is required, such as bent caps in bridges or coupling beams in high-rise buildings [1,2]. In conformance with ACI 318 definition [3], an RC deep beam's clear span should be equal to or less than four times its depth. Besides, the concentrated load must be within twice the member depth, measured from the member end [4]. From the failure patterns standpoint, the RC deep beam is considered one of the most complex structures, and generally, failure occurs due to the brittle shear effect [5]. Due to uncertainty, many studies have been conducted to observe its failure mechanism and predict its ultimate strength [6–8]. A

web opening in RC deep beams is sometimes essential because it utilizes ducts and pipes and accommodates services, including electricity supplies, gas, and water [9,10]. These web openings can distort the RC deep beam's stress distributions and consequently affect the structural behavior, which requires more attention to reinforcement details or CFRP reinforcement [11,12].

The properties of CFRP, a polymer matrix reinforced with carbon fibers, can be summarized by the outstanding modulus of elasticity, high tensile strength equals, low-density equals, and excellent chemical inertness [13]. CFRP is expensive, but it has very high mechanical properties; however, one of the main disadvantages is brittleness [13]. Therefore, the primary use of CFRP is for reinforcing and rehabilitating a concrete member subjected to tensile stresses. Several studies have investigated CFRP properties to understand its behaviors and explore its applications. For instance, Zhang et al. [14] elaborated on mitigating blasts on structures. Wide strain ranges were applied to investigate the unidirectional tensile properties, and it was found that the tensile strength for CFRP/epoxy laminates was not affected by loading speed. A model has been introduced to understand 3-D viscoelastic characterizations and the effect of time and temperature dependence, and promising results were achieved [15]. Li et al. [16] investigated the effect of incorporating the vapor-grown carbon fiber (VGCF) powder method on the mechanical properties of CFRP laminates. The results showed that hybrid CFRP/VGCF laminates improved fracture behavior and crack propagation. It was found that the moisture ingress with high temperature would significantly decay CFRP's mechanical properties; hence, it was proposed to a hot/wet design to mitigate the effect of humidity while the high temperature is presented [17]. Moon et al. [18] found that when multi-wall carbon nanotubes (MWCNT) thin-ply were added to CFRP reinforced thin-ply, the tensile strength properties were improved at ground conditions and the degradation rate was deaccelerated remarkably at low earth orbit (LEO) environments.

Several applications and studies on CFRP are being used for reinforced concrete structures, including but not limited to all types of regular concrete beams and deep beams. Bousselham and Chaallal [19] experimentally investigated the performance of reinforced concrete T-beams in shear reinforced with CFRP. It was found that CFRP thickness was not proportional to the shear resistance, while the resistance depended on internal transverse steel reinforcements. Abduljalil [20] studied the effect of bonded CFRP stirrups and fiber orientation on the flexural and shear strength of a deep beam with openings. The experimental results showed that CFRP with 45° orientation sustained higher cracking and ultimate load than the one with 90° orientation. The deep beam ductility was improved, and it was found that combining the aforementioned orientations would further increase the cracking and ultimate loads [20]. Lee et al. [21] investigated the performance of deep beams reinforced with CFRP. It was reported that strengthening length, anchorage, and fiber direction were considered the most prominent vital factors affecting the deep beam shear performance.

Strut-and-tie modeling (STM) is a unified approach that mirrors all load influences, including moment, axial and shear load, and tension force. Several studies showed that STM is an advantageous method for designing RC deep beams and predicting their structural behaviors [22–24]. The STM introduces a rational approach by analyzing a complex element into simplified truss models, as shown in Figure 1 [25]. In the STM approach, every design case study is unique and not similar to another; however, some rules and techniques help structural designers develop and solve these case studies [26]. This study will use the STM approach to explain and justify the RC deep beam behaviors and influence of CFRP when openings are introduced to the structural member.

A finite element (FE) model has been proposed by Hawileh et al. [27] to simulate the performance of reinforced concrete deep beams strengthened externally with CFRP plates. The model revealed a satisfactory agreement with the collected experimental data, and it can accurately predict a debonding failure. Based on an extension of the STM, a simple procedure was introduced and validated to predict the shear capacity of a reinforced concrete deep beam strengthened with CFRP sheets [25]. Hanoon et al. [28]

proposed a new effectiveness factor depending on STM to estimate shear strength for deep beams strengthened with CFRP. The model showed a satisfactory result in predicting shear strength and its failure modes, such as concrete crushing or diagonal splitting. As aforementioned in the literature, there is a lack of literature regarding the effect of opening size on the strut-tie model behaviors in the deep beam and the effect of CFRP on the stress distribution in the doubly/singly reinforced deep beam.

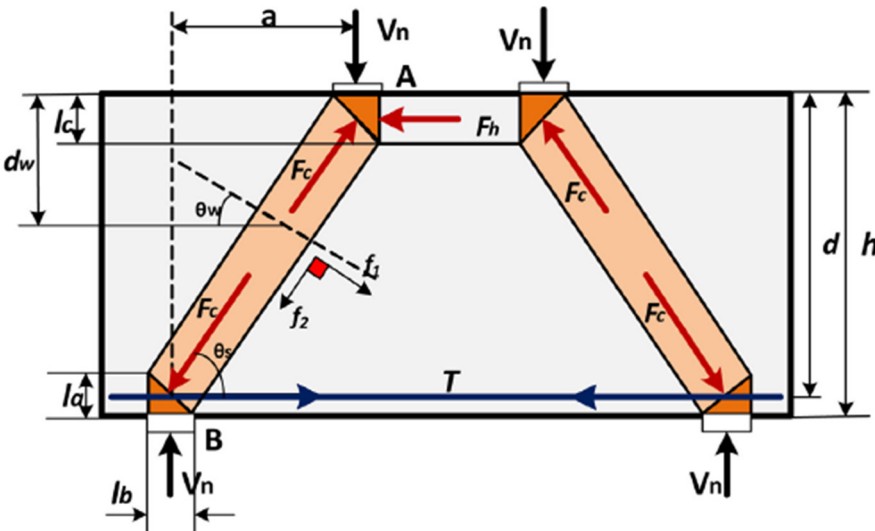

**Figure 1.** Strut-and-tie Modeling [25]. $V_n$ = reactions; $F_c$ = compression load; T = tension load.

After ascertaining the finite element approach capability, a parametric study was imposed and investigated, which was used to evaluate the influence of several opening sizes with CFRP reinforcement using the STM concept, structural capacity, and performance of the member and the resulting stress distribution of the RC deep beam.

## 2. Research Objectives

The main prepossesses of the paper can be listed as follows:

A.   Investigate different CFRP strengthening solutions for singly/ doubly RC deep beams on the STM stress web opening.
B.   Analyze the effectiveness of the suggested strengthening solutions to restore the beams' shear resistance through a comparison between laboratory testing and the finite element models, then verify numerical predictions.
C.   Investigate the effect of opening size for doubly/singly reinforced concrete deep beams on STM stress behavior and CFRP's effect on the capacity of the structure and stress–strain distributions.

## 3. Model Development and Validation

Validation of the FE against a selected experimental work and a parametric study were conducted in the current study. The parametric study evaluated the influence of several opening sizes with/without CFRP on the STM, the load-deflection response, and the stress distribution of the RC deep beam. The finite element model, used in the study, has been validated in several previously published works [29–31].

### 3.1. Experiment Description

To validate the ability of the selected concrete model for studying the behavior of deep beams, a validating model was created and compared with an experimental study [32]. Three selected experimental cases, a deep beam with and without an opening and an RC deep beam with opening plus CFRP stirrups [32], were selected to validate the RC model behaviors and results. This test serves as a source for comparison between the experi-

mental results and the numerical analysis. In the current study conducted by Nadir [32], simply supported deep beams were instrumented with devices to measure the mid-span deflections and loads. Figure 2 illustrates the cross-section and loading configuration of the experimentally tested beams.

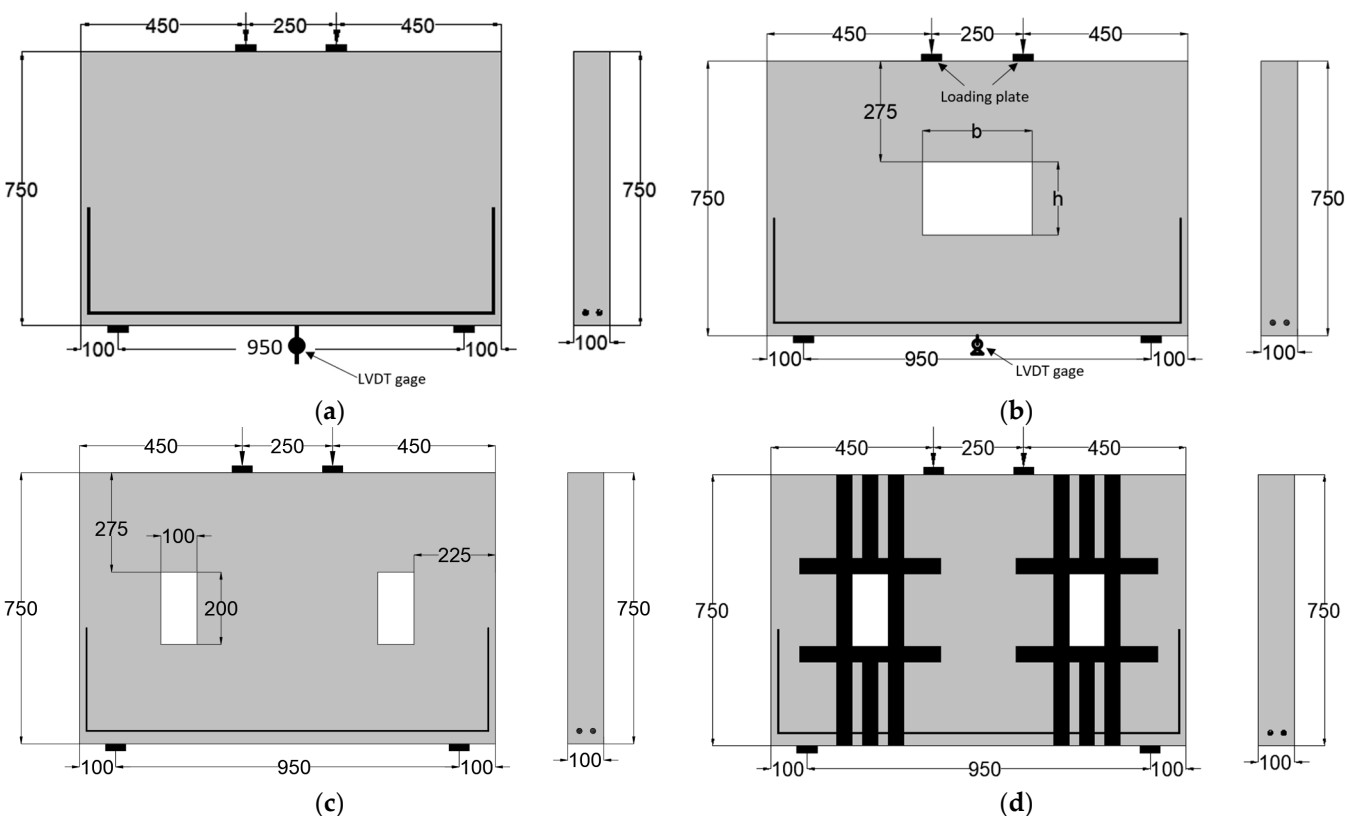

**Figure 2.** Cross-section and loading configuration for the experimental specimens. (**a**) Control sample details; (**b**) RC deep beam with one web opening at the center; (**c**) RC deep beam with two web openings at the center; (**d**) CFRP configuration details.

### 3.2. Finite Element Description

All the deep beam specimens were modeled in ABAQUS software. The damaged plasticity model in the material library of the finite element ABAQUS software was adopted to model the concrete material. The concrete was modeled with an 8-noded hexahedral (brick) element with reduced integration (known as C3D8R). To represent internal steel reinforcement, a two-node linear displacement truss element (denoted as T3D2) was chosen. A perfect bond was assumed by employing the embedded strategy between the grid reinforcement and the surrounding concrete material. This was considered reasonable because ribbed and bent bars were used in the experimental tests conducted by Nadir [32]. In addition, the truss elements are used in two and three dimensions to model slender, line-like structures that support loading only along the element's axis or centerline. The two-dimensional truss elements can be used in axisymmetric models to represent components, such as bolts or connectors [33]. Several studies [29–31,34] have adopted this mathematical representation to represent the steel reinforcement due to the identical representation of obtained results from the mathematical model compared to laboratory samples. The nodes at the supports were restricted against vertical displacement. Based on several iterations with different sizes, the optimum element size used in the modeling was 20 mm. The bonding between concrete and CFRP sheets was assumed to be fully bonded using the tie constrain technique in the Abaqus software. All the samples were loaded to failure by displacement control (four-point displacement load) in the vertical direction from the

loading point located above the beam and connected to the beam loading surface. The meshed models used in this validation are shown in Figure 3.

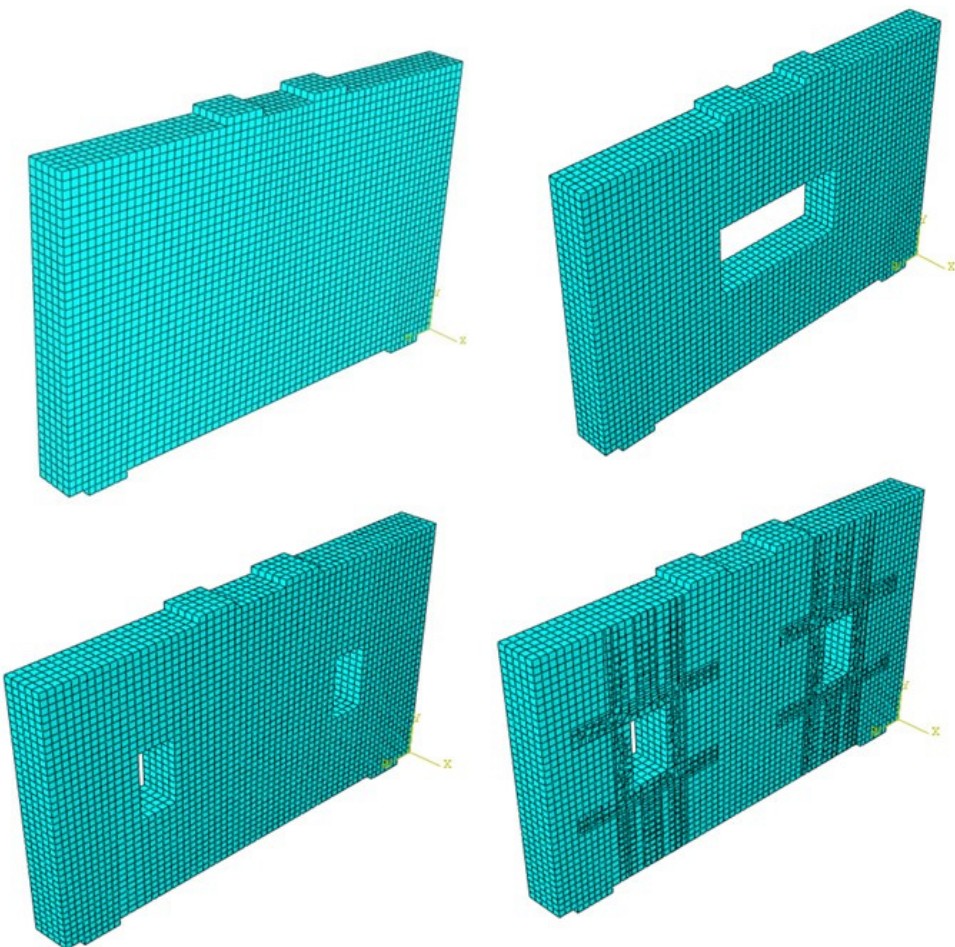

**Figure 3.** Three-dimensional finite element model.

The modeled response was set to verify the ability of the selected model to capture the whole beam's behavior up to failure and the influence of CFRP on structural performance. The model results can be used to validate and guide more numerical analysis investigation and explore concrete response under complicated loading conditions, such as the behaviors of reinforced concrete deep beams with/without web-opening.

The mechanical properties of concrete used in the numerical analysis were based on Nadir's experimental study [32]. Table 1 enlists the properties of concrete, steel reinforcement, and carbon fiber-reinforced polymer (CFRP) sheets used to strengthen the openings.

**Table 1.** Properties of the used materials.

| Steel Reinforcement | | Concrete | |
|---|---|---|---|
| Yield strength (MPa) | 540 | Compressive strength (MPa) | 27.15 |
| Modulus of elasticity (MPa) | $2 \times 10^5$ | Modulus of elasticity (MPa) | $24.5 \times 10^3$ |
| Ultimate strength (MPa) | 640 | Poison ratio | 0.2 |
| Poison ratio | 0.3 | Dilation angle | 37 |
| **CFRP Sheet** | | Eccentricity | 0.1 |
| Tensile strength (MPa) | 4400 | $f_{bo}/f_{co}$ | 1.16 |
| Modulus of elasticity (MPa) | $2.38 \times 10^5$ | $K$ | 0.667 |
| Elongation at break (%) | 1.8 | Viscosity | 0 [33] |
| Thickness (mm) | 0.131 | | |

A nonlinear elastic behavior was selected for the steel reinforcement bars. Table 1 illustrates the properties of the material. More information regarding model simulations and theoretical assumptions can be found in several papers [35,36].

### 3.3. Validation of the FE Models

According to Nadir's experimental work, Figure 4 shows the failures of different deep beam configuration specimens, including a solid deep beam, a deep beam with two web openings, and a deep beam with two web openings strengthened by O-shape-CFRP stirrups around the opening. The failure mode of the solid deep beam initiated with a small flexural crack of concrete at the tension part of the model after achieving 30% of the peak load, which was seen in a lower load value than that the visual crack appeared in the experimental test. As shown in Figure 4a, the strain distribution coming from FE analyses depicts the failure pattern that occurred in the experiment. The failure is due to inclined shear stress within the strut. The failure started from the applied load toward the supports. Regarding the model with two openings, the failure mode started with a diagonal crack at the lower corner of the openings with minimal flexural stress in the tension region that appeared before the next loading stage. Then, the development of inclined stress in the opposite corner led the specimen to fail with an ultimate load of approximately 163 kN, as shown in Figure 4b.

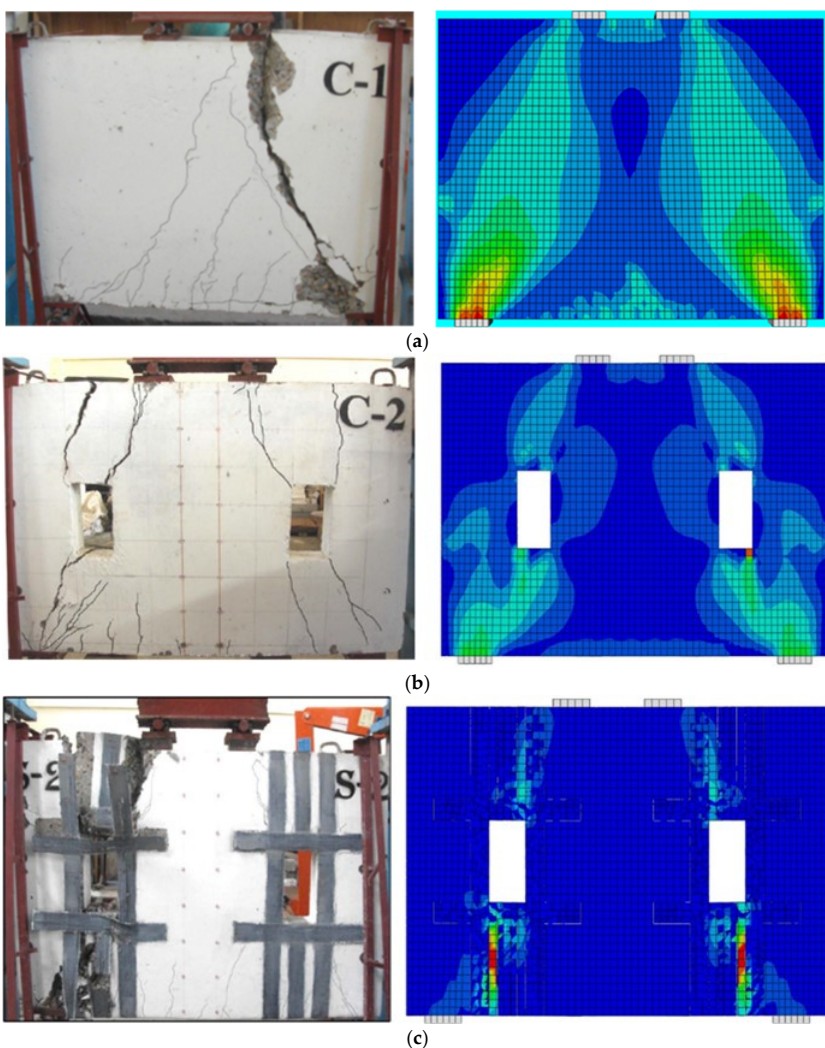

**Figure 4.** Comparison of experimental and numerical damages and behaviors: (**a**) control specimen vs model; (**b**) RC two web openings sample vs model; (**c**) RC deep beam with two web openings +CFRP vs model.

Regarding the specimen with two openings, strengthened with O-CFRP stirrups wrapping around the opening, the first strain distribution appeared at the adjacent sides of the opening when the specimen's deflection reached 0.5 mm in the FE simulation. Simultaneously, the crack was visible in the experimental test when the deflection was about 0.35 mm, which may be attributed to the assumption of full bonding between the concrete and CFRP stirrups. It is followed by compression stress concentration between the region between perpendicular CFRP stirrups. As a result, the specimen failed by splitting concrete in the corners above and below the opening, as shown in Figure 4c. The damaged regions mainly occurred at the strut region between the loading and supporting plates as well as at the diagonally mirrored corners of web-opening across the compression strut. Consequently, it can be distinctly observed that the simulated configurations of failure modes in the FE model are consistent with the experimental findings.

Figure 5 compares the load-deflection curves of the modeled members versus the corresponding experimental ones. It is observed that the predicted load-deflection curve is relatively similar to the experimental curves. It concludes that the chosen FE model and inputs are capable of providing reliable results and can be further used to evaluate the failure mechanism of deep beam specimens (with/without CFRP strengthening) under a four-point bending load.

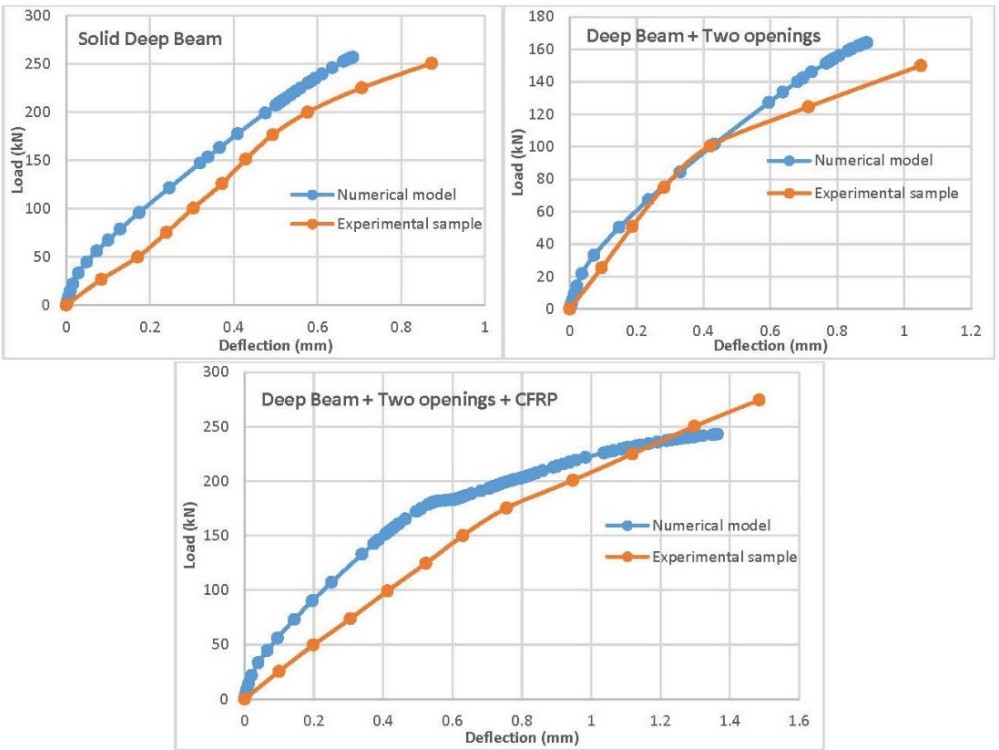

**Figure 5.** Comparison of load and mid-span deflection curve for solid specimen, two openings, and CFRP specimens.

## 4. Parametric Studies

### 4.1. General Description

For parametric studies, the FE model was utilized to investigate the influence of several size openings with/without CFRP on the behavior of deep beams for four groups. Nine deep beams of the same geometrical dimensions for each group. Different opening sizes with and without CFRP were modeled and analyzed. The reported material properties in Table 1 were adopted for the parametric studies.

The parametric studies in this section aim to provide an extensive understanding of achieving a better reinforcement configuration, opening ratio, and strengthening effect. Besides, it will assist with observing the effect of CFRP-stirrup strengthening of deep beam

specimens with the various web-opening ratio and related parameters. The parameters of interest include web-opening ratio, longitudinal reinforcement ratio, and CFRP stirrups surrounding the openings. All specimens have simply supported boundary conditions. They have been categorized into four main groups with the same compressive strength and opening ratio. The first group is eight specimens with different web-opening ratios (b/h) and three configurations with 2Ø16 longitudinal reinforcement (beam DA100-DA800). The second group (Group B) includes eight specimens with a web-opening ratio similar to group A except for 2Ø16 longitudinal reinforcements in the compression zone (beam DB100–DB800). The third group has eight specimens with the same web-opening configuration as in groups A and B, but a 2Ø16 longitudinal web reinforcement bar. Table 2 shows the details of the opening ratio and the reinforcement configuration for the three groups A, B, and C. On the other hand, the fourth group (Group D) contains thirteen specimens, consisting of five beams with a web-opening ratio of (200/700 mm/mm) and eight beams with a web-opening ratio of (200/800 mm/mm) as summarized in Tables 2 and 3. Figure 6 illustrates the reinforcement details of the Groups A, B, and C. Figure 7 shows the CFRP configurations for Group D. Four-point bending was performed (two concentrated loads above the specimens).

**Table 2.** Details of opening ratio and reinforcement for specimens in groups A, B, and C.

| Group Name | Opening Ratio b/h (mm/mm) | | | | | | | |
|---|---|---|---|---|---|---|---|---|
| | 100/200 | 200/200 | 300/200 | 400/200 | 500/200 | 600/200 | 700/200 | 800/200 |
| | 0.5 | 1 | 1.5 | 2 | 2.5 | 3 | 3.5 | 4 |
| Group A | DA100 | DA200 | DA300 | DA400 | DA500 | DA600 | DA700 | DA800 |
| Group B | DB100 | DB200 | DB300 | DB400 | DB500 | DB600 | DB700 | DB800 |
| Group C | DC100 | DC200 | DC300 | DC400 | DC500 | DC600 | DC700 | DC800 |

| | Longitudinal reinforcement | | | |
|---|---|---|---|---|
| | Tension zone | Compression zone | Web zone | - |
| Group A | 2Ø16 | - | - | - |
| Group B | 2Ø16 | 2Ø16 | - | - |
| Group C | 2Ø16 | 2Ø16 | 2Ø16 | - |

Note: DA100 stands for D: deep beam, A: group A, and 100: opening width in mm.

**Table 3.** Details of fourth group D [1] strengthened with CFRP.

| Model no. | Opening Ratio (*h/b*) | Strengthening Technique with CFRP | | CFRP-Stirrup Width (mm) |
|---|---|---|---|---|
| | | CFRP-Straight Sheet | | |
| | | Length (mm) | Width (mm) | |
| DS-700-0 | | | | without |
| DS-700-100 | | | | 100 |
| DS-700-120 | 200/700 | 700 | 100 | 120 |
| DS-700-140 | | | | 140 |
| DS-700-160 | | | | 160 |
| DS-800-0 | | | | without |
| DS-800-100 | | | | 100 |
| DS-800-120 | | | | 120 |
| DS-800-140 | | | | 140 |
| DS-800-160 | 200/800 | 800 | 100 | 160 |
| DS-800-180 | | | | 180 |
| DS-800-200 | | | | 200 |
| DS-800-220 | | | | 220 |

Note: [1] = the longitudinal reinforcement of the fourth group is similar to the third group, and the thickness of the CFRP sheet and stirrups are 0.131 mm.

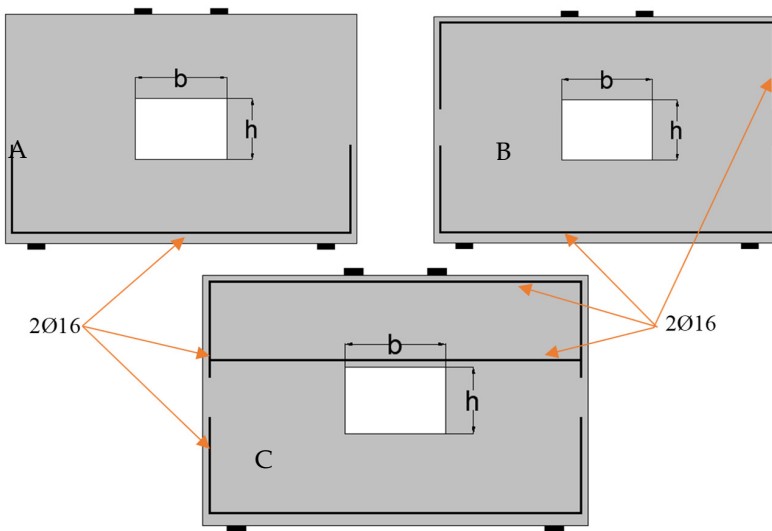

**Figure 6.** Reinforcement details of groups A, B, and C.

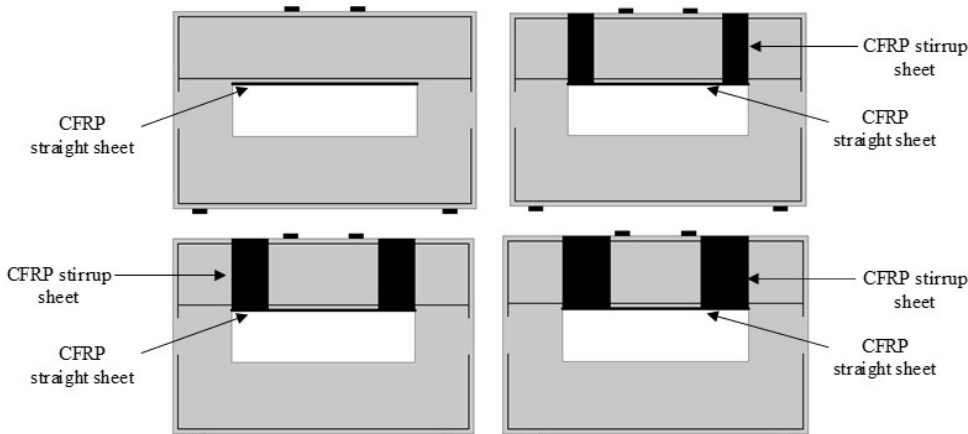

**Figure 7.** Details of reinforcement with opening at the center and strengthening pattern of group D.

### 4.2. Results and Discussions

#### 4.2.1. Effect of Opening Ratio

The observations of ultimate load capacity at the different opening ratios for the deep beam specimens in group A showed that the ultimate load of the solid beam is significantly more than that of the model with a web-opening, DA800, by 71%. The rational reason behind that is a lack in the beam web's concrete mass resulting in a drop in the deep beam stiffness and its flexural strength. As shown in Figure 8, the ultimate load capacity of the solid deep beam is 256.8 kN, which is greater in comparison to the ultimate load capacity of other specimens DA100, DA200, DA300, and DA400, which is 250.6, 241.7, 230.3, and 216.7 kN, respectively. By considering the solid model result as a reference, it can be seen that the reduction in ultimate load capacity for the specimens with a web-opening ratio (b/h) of 0.5, 1.0, 1.5, and 2.0 (relative to the solid specimen) was 2.4%, 6.0%, 10.3%, and 15.6%, respectively. Furthermore, it was more obvious when the opening ratio (b/h) increased by more than 2.0. For instance, the ultimate load capacity of the specimens with (b/h) ratio of 2.5, 3.0, 3.5, and 4.0 (DA500, DA600, DA700, and DA800) was 171.0, 124.0, 107.0, and 75.0 kN, respectively. The reduction percentage compared to the solid model is 33.1%, 47.6%, 58.1%, and 71.0%, respectively. The drawn curve of load versus (b/h) ratio appeared to be bilinear, in which the first segment started from 0 to 2.0, while the other segment started from 2.0 to 4.0.

As shown in Figure 9, it helped to justify these two different behaviors. It can be postulated that the opening disrupted the strut when the width of the opening became

more than 400 mm. Crossing the compression struts, developed between the applied load and the supports for the case of specimens DA400, DA500, DA600, and DA700, resulted in a substantial reduction in the structural performance and overall beam capacity, as depicted in Figure 9a,b. Therefore, the main observation is that if the opening width does not interfere with the load path and stress trajectories, i.e., compression struts, the overall performance of opening size is not predominant. In other words, once the percentage of opening width (b) over a distance between deep beam supports (L) is within 42%, structurally, there are no pronounced effects. Increasing the web-opening ratio (b/h) by more than 2.0 results in a shift in the failure path, leading to the failure stresses concentrated in the upper part of the web-opening, as shown in Figure 9b. Figure 10 illustrates the strain distribution for solid beams and beams with different opening web ratios and the same opening depth with various lengths. As shown in Figure 10, it can be seen from the models (DA0-DA400) that the compression strut has developed between the load and supports for specimens with a b/h ratio less than 2.0. The compression stresses, and compression strut, started to shift their direction and narrow down when the width of the opening began to widen, which was apparent in the model (DA400). The reduction in ultimate load capacity was no more than 15%, as shown in Figure 8.

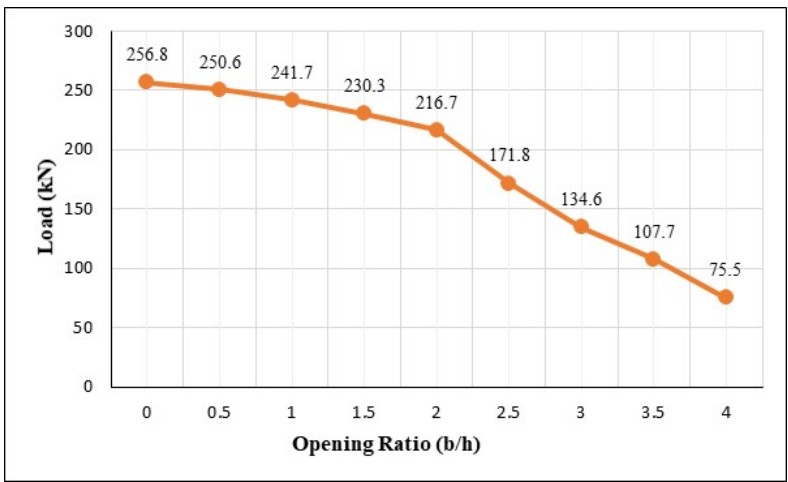

**Figure 8.** Load verse opening ratio of group A.

On the other hand, cracking stress failure of specimens DA500 and DA600 was initiated in the upper part between the loading plate and the upper corners of the web opening. Finally, the specimens, DA700 and DA800, were sustained from flexural cracks that emerged in the tension zone area at the upper part of the openings. Due to the increase of the web-opening ratio by more than 3.5%, the specimens (DA700 and DA800) acted as two separate beams instead of single deep beam behaviors. It resulted in a rapid reduction in the overall capacity of the specimens in comparison with specimens (DA100, DA200, DA300, and DA400).

In conclusion, the STM theory and analyses will not be applicable when the opening ratio (b/h) and length-to-web ratio (b/L) increase by more than 200% and 42%, respectively. The member will behave as two separate flexural members. While the upper member, which is located over the openings, sustained the most strain distribution, the lower member seems intact due to stress disruptions by openings.

### 4.2.2. Effect of Reinforcement Detailing

Because of increasing the web-opening size in the deep beam, the overall capacity of members has been reduced significantly. The members with a wide web opening, such as 500, 600, 700, and 800 mm, have behaved as two separate flexural beam members rather than one full deep beam member, as shown in Figure 8. As aforementioned, in group B, in addition to longitudinal reinforcement steel bars provided in the tension zone, such as in

group A, a 2Ø16 longitudinal reinforcement bar was added to the compression zone, as shown in Figure 6 and Table 2. In group C, 2Ø16 longitudinal bars were used to reinforce the upper part of the opening in addition to the 2Ø16 longitudinal bars in the compression and tension zone, respectively. The results showed no significant contribution to utilizing the longitudinal reinforcement in the compression zone for group B. The deformation and failure mode of most samples did not alter the results compared to the members of group A. Figure 11 presents the maximum applied load versus opening ratio (b/h). It can be seen that the failure load of groups A and B are almost identical. The outcomes were expected because the reinforcement detail in group B did not mitigate the main issue, which has a redistributed tension stress in the upper part of the opening. As shown in Figure 12, the strain distribution of group B is similar to the strain distribution of group A. This mechanism of failure occurred due to the stress concentration at the mid-span above the web opening leading the members to fail in a flexural manner, with a lower value of ultimate load capacity in comparison with the member without web-opening.

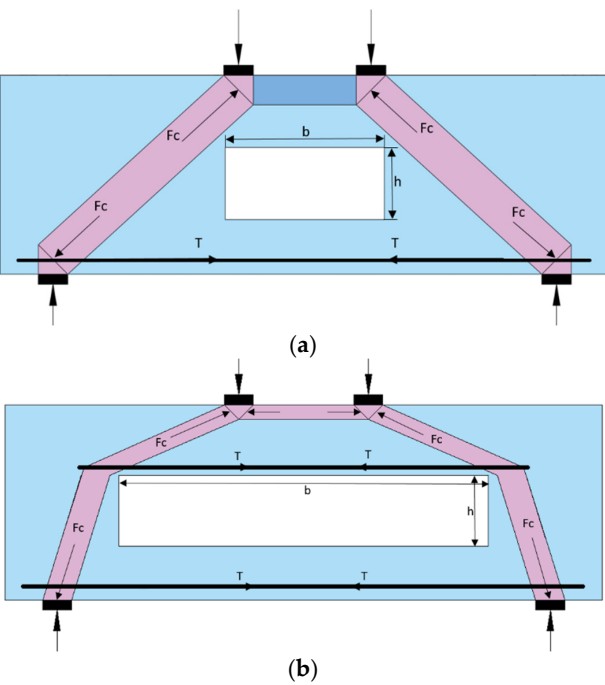

(a)

(b)

**Figure 9.** Effect of opening ratio on the STM; (**a**) b/h ≥ 2, (**b**) b/h < 2.

Regarding group C, it appeared that incorporating longitudinal reinforcement above the web-opening of the members effectively increased the ultimate load capacity compared to groups A and B, as shown in Figure 11. The most notable improvement in ultimate load capacity was detected in the DC400 and DC800 members. The increase in the overall capacity of the members DC400 and DC800 was 10% and 35%, respectively, in comparison to group A. Therefore, it can conclude that the presence of the web reinforcement was more prominent in minimizing the propagation of cracks that were dominant in group A and B; particularly at the tension zone above the web-opening, as shown in Figure 13. Figure 11 shows that the STM has an extensive effect on the b/h ratio beyond 200%. For specimens with a b/h ratio in a range of 200–300%, stress concentration at the tension zone above the web opening has been altered and redistributed to the support through the compression strut due to web rebars' contribution by resisting tension stress propagation in comparison with group A and B results. However, diagonal cracks have been noticed in specimens DC700 and DC800, which indicates that the failure mechanism of DC700 and DC800 is due to flexural failure in the upper part of the beam. The deep beam acted as two separate beams, and the compression strut disappeared gradually, similar to the A and B groups.

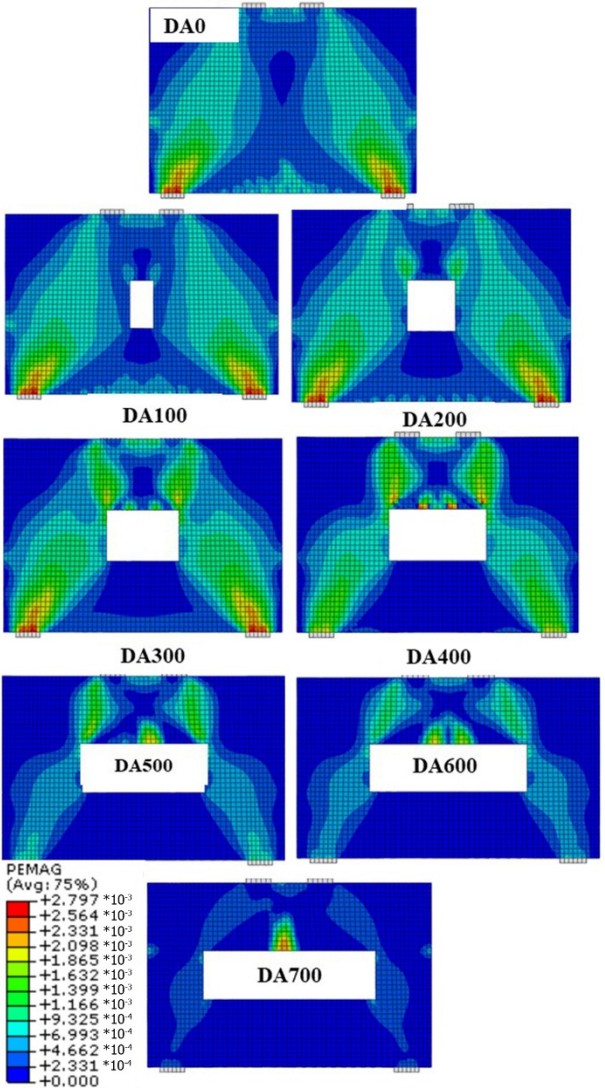

**Figure 10.** Strain distribution of group A.

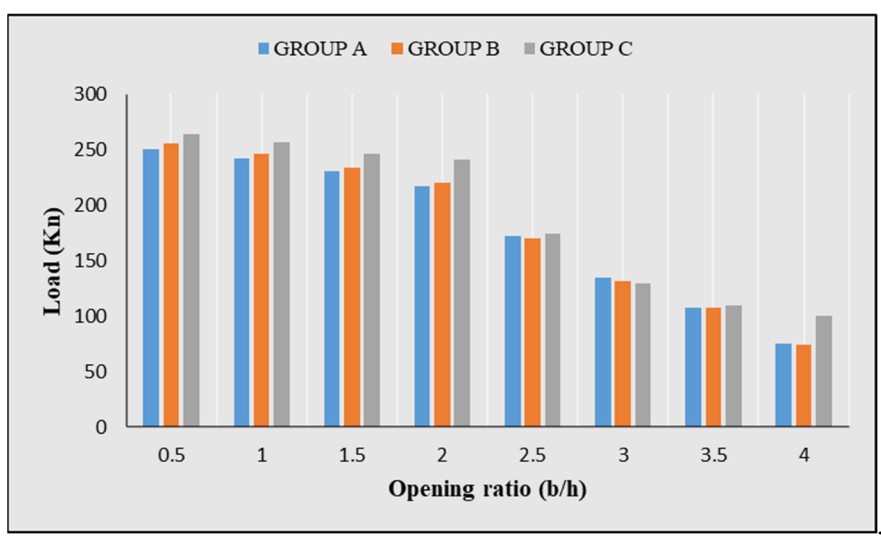

**Figure 11.** Load verse opening ratio for groups A, B, and C.

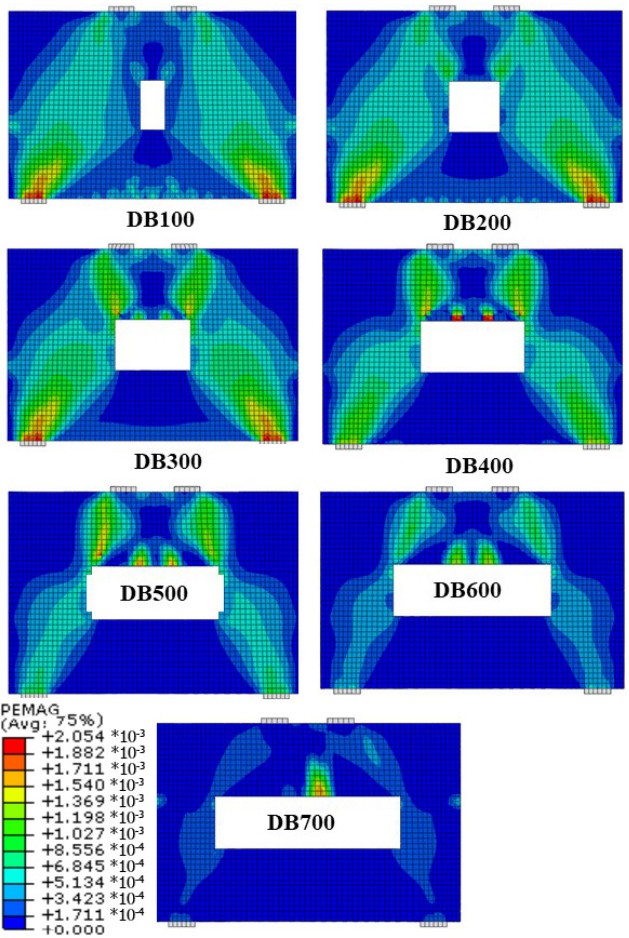

**Figure 12.** Strain distribution of group B.

### 4.2.3. Effect of CFRP

As aforementioned in Section 3.2, ABAQUS software was used for simulating the behavior of deep beams strengthened with CFRP for investigating the effect of CFRP reinforcement. A CFRP strengthening technique has been used in a deep beam with a large web openings ratio, i.e., DB700 and DB800. As shown in Figure 6, a straight one-layer with varying stirrup widths of CFRP has been applied to the region above the web-opening specimens. The reason for choosing the opening with 700 and 800 mm is that these two specimens showed the lowest strength. It is also to observe the effects of CFRP in providing additional strength and redistributing the strain concentration from the upper part of the member to the supports.

Figure 14 compares the ultimate load of DB700 and DB800 models reinforced with 0, 100, 120, 140, 160, and 180 mm CFRP stirrups, as summarized in Table 3. The results showed no apparent effect for CFRP-stirrup when the width was between 0 mm to 140 mm. Then, the results of the DS-700-160 revealed that when the width of CFRP-stirrups increased from 140 to 160 mm, the ultimate load capacity value of the members was improved. This is because the strengthening with one sheet and different stirrup widths of CFRP in the web opening not only minimized the diagonal cracks of the original members but also changed its failure mode, which improved the overall deformation ability of the members and resulted in a significant increase in ultimate load capacity, as depicted in Figure 15. The results showed that the optimum CFRP stirrup width should be over 150 mm for a deep beam with a 700 mm opening width.

The same results pattern was observed in the case of the specimens with an 800 mm opening width, as shown in Figure 16. For instance, when 0–140 mm width CFRP-stirrup was used, a minimal increase in the ultimate load capacity of group D with an 800 mm web

opening was observed. However, a noticeable enhancement in overall capacity occurred while using a CFRP stirrup width between 160 mm to 200 mm. The postulated reason is that having a CFRP stirrup with enough width helps control the propagation of diagonal cracks on both sides of the web opening and confinement to the upper part of the deep beam.

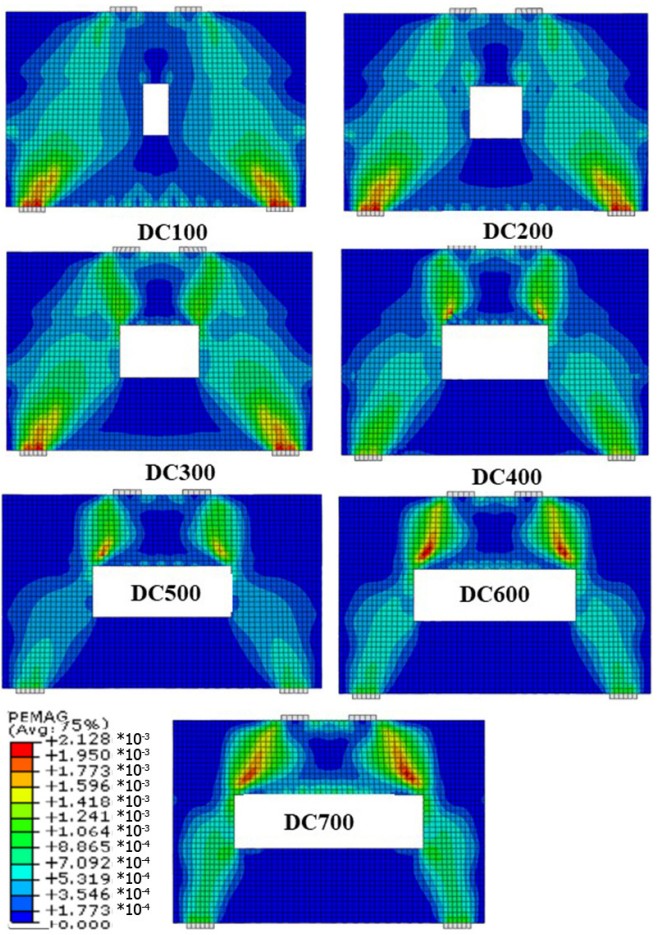

**Figure 13.** Development of compression struts for group C.

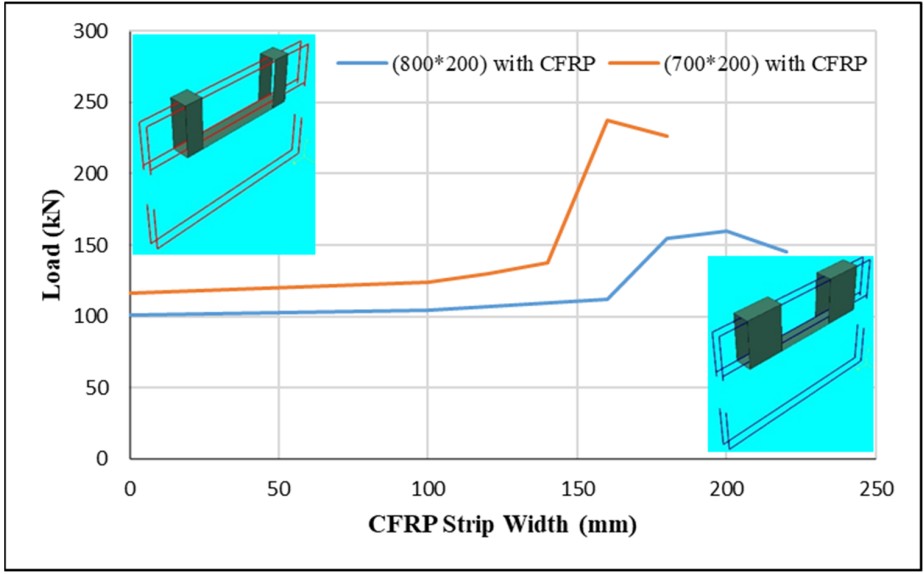

**Figure 14.** Ultimate load vs CFRP stirrup width.

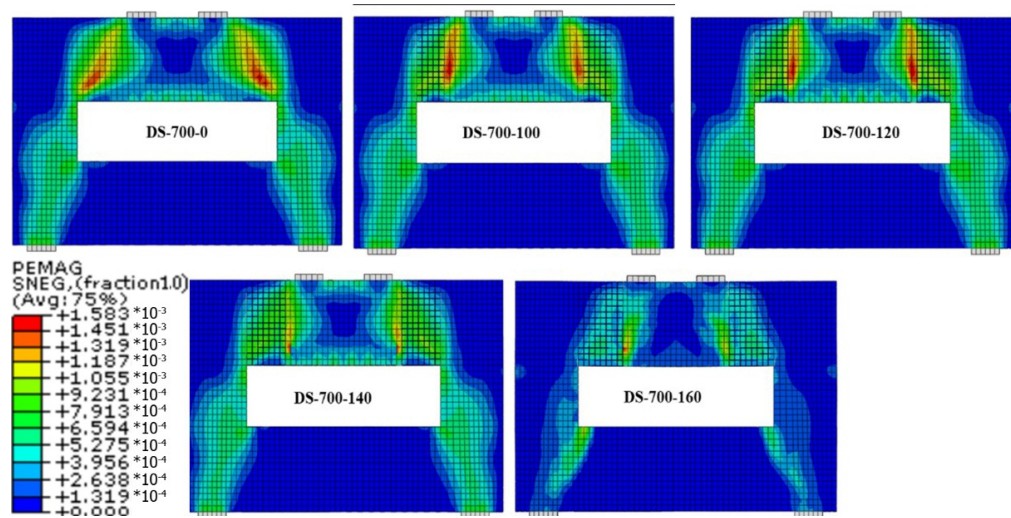

**Figure 15.** Effect of CFRP on strain distribution of DS-700.

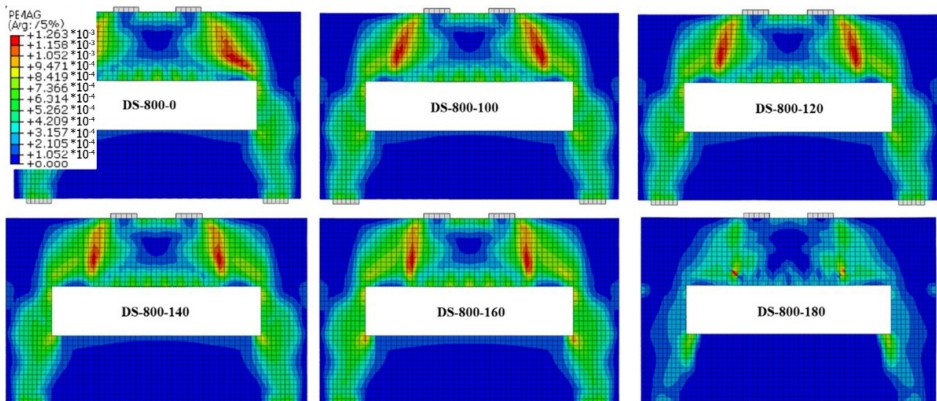

**Figure 16.** Effect of CFRP on strain distribution of DS-800.

Finally, CFRP-sheet strengthening in the top face and CFRP-stirrups at the top region of the web-opening of RC deep beams has remarkably increased the beam strength, as shown in Figure 17. The maximum strength gain was 57% and 117% when the web-opening size was 800 × 200 and 700 × 200 mm, respectively, where most of the shear force was carried by the top chord that had wrapped with CFRP stirrup on both sides.

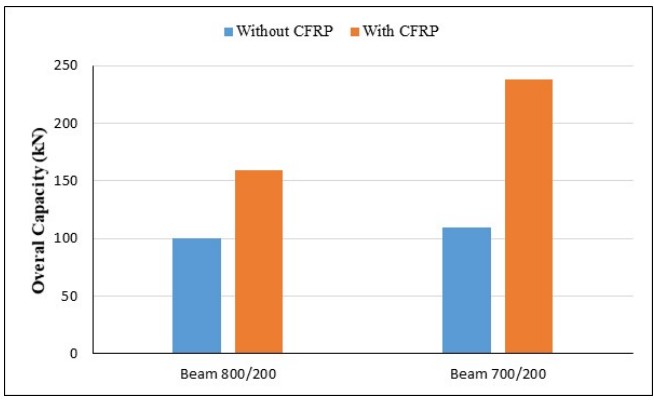

**Figure 17.** Comparison of deep beam with and without CFRP.

## 5. Application of RC Deep Beam in Infrastructure

The applications of RC deep beams in infrastructures are varied, such as buildings, bridges, pile caps, transfer girders, shear walls, etc. RC deep beams are a very useful structural system; however, its failure pattern is complicated due to intense shear stresses. In addition, the presence of web-opening in the RC deep beam is essential to increase its efficiency in providing the service because there is a need for pipes and electricity appliances to pass through the RC deep beam. There are no particular standards/codes for their size and shapes. However, ACI 318 definition [3], an RC deep beam's clear span should be equal to or less than four times its depth. Besides, the concentrated load must be within twice the member depth, measured from the member end [4]. Therefore, the primary reason for the RC concrete deep beam failure is high shear stress. The CFRP usage in some infrastructure members becomes common due to availability and friendly installations. Many infrastructures' rehabilitation and enhancement, such as beams or columns, are improved or repaired when CFRP is being used.

## 6. Conclusions

This study validated an experimental result of reinforced deep beams with and without CFRP and a numerical result using ABAQUS software. The parametric studies investigated the effects of opening ratio (b/h), reinforcement details, and CFRP stirrups width on reinforced deep beam performance with several opening sizes. The research concluded several points as follows:

The performance and strain distribution of the deep beam were significantly impacted by the opening size ratio (b/h). The failure load was substantially reduced when the b/h ratio changed from 0 to 400%. The strain distribution and failure mode changed from a reinforced deep beam with one single STM unit to two separate members. In the case of two separate members, the upper member sustained the main load and the resulting stresses, and the lower member did not have any damage approximately.

Once the opening ratio (b/h) and length-to-web ratio (b/L) increase by more than 200% and 42%, respectively, special attention should be paid to the upper part, which is over the web opening, because the main stresses will be concentrated there. If performance improvement is the main target, the upper part of the RC deep beam should be reinforced with a web reinforcement, and confinement by using CFRP would be a plus.

The reinforcement in the compression zone did not influence the reinforced concrete deep beam with a web opening. However, when the steel reinforcement was used to reinforce the upper part of the opening, a better performance was noticed in terms of stress distribution. For example, by comparing the results of group A, which only had reinforcement in the tension zone, and the result of group C, which also had reinforcement on the web opening, the failure load improved by 35%.

CFRP stirrup with a width from 0 to 140 mm did not improve the load capacity of the reinforced concrete deep beam with web-opening; however, noticeable enhancement was observed when wider CFRP stirrups were used.

CFRP stirrups utilizations are important in infrastructure rehabilitation, such as bridges, beams, and column damage, because they are easy to install and can significantly improve member performance. The results showed that CFRP improved the capacity of the RC deep beam effectively even though the size of the web opening was increased.

An RC deep beam with its opening reinforced with CFRP stirrups has many applications, such as bridges and buildings. Increasing the size of the opening can alter the stress distribution in the RC deep beam; therefore, it is essential to understand the stress redistribution and tackle the issue by using the CFRP stirrups. When an RC deep beam with an opening fails due to increasing the applied load, a CFRP is a viable practical solution because it is easy to install and maintain.

**Author Contributions:** Conceptualisation, Y.A.A. and L.N.A.; methodology, Y.A.A. and L.N.A.; validation, L.N.A. and S.G.; formal analysis, Y.A.A.; investigation, H.R.T. and H.A.; data curation,

L.N.A.; writing—original draft preparation, Y.A.A.; writing—review and editing, Y.A.A. and H.A.; supervision, L.N.A. and C.N.D. All authors have read and agreed to the published version of the manuscript.

**Funding:** This research received no external funding.

**Data Availability Statement:** Not applicable.

**Conflicts of Interest:** The authors declare no conflict of interest.

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
