# Peer review of "Numerical Investigation on Effect of Opening Ratio on Structural Performance of Reinforced Concrete Deep Beam Reinforced with CFRP Enhancements"

_infrastructures, doi:10.3390/infrastructures8010002_

Round 1

Reviewer 1 Report

 on the whole,this paper is a good paper.but also want to present some advices:

1 it need minor spell check;

2 some references's sources are not clear. for example 2

3 the value of all the strain cloud cannot be seen clearly,so the reader cannot get more information  

4 if the columns in figure10 were drawn in different color,that is ok.but now group B and group C cannot be seperated clearly.  

5 if there are more analysis on the load bear capacity of all the specimens, I think the paper will be better than this one

Author Response

The authors wish to thank the reviewer for the comments and for the timely and thoughtful review. Each comment is addressed individually below.

Comments and Suggestions for Authors

On the whole, this paper is a good paper. but also want to present some advices:

1 It need minor spell check.

Response. Agree. The authors conducted a rigor spell, grammatical, punctuation check.  

2 Some references's sources are not clear. for example 2

Response. Agree. The authors corrected the mentioned reference and check the others.

3 The value of all the strain cloud cannot be seen clearly, so the reader cannot get more information  

Response. Agree. The authors used new high-resolution figures.

4 If the columns in figure10 were drawn in different color, that is ok.but now group B and group C cannot be seperated clearly.

Response. Agree. Authors elaborated on the comments and modified the mentioned figures.

5 If there are more analysis on the load bear capacity of all the specimens, I think the paper will be better than this one.

Response. Agree. Due to the time number of page limitations, the authors decided to skip the analyses.

Reviewer 2 Report

This topic is quite good, discussing "the behavior of simply supported concrete deep beams reinforced with carbon fiber-reinforced polymer (CFRP)". However, the publication still needs some improvements, including:

Line 19-20, Introduction to the abstract needs consideration for improvement, is it true that roadway culverts, and foundation pile caps have the same behavior as deep beams? If we look at the field the position and loading is very different.

Line 36, In the abstract, please give advice on the effective use of hollow deep beams in the field according to the results of your research

Line 188, what is “Figure 0”, please check.

Line 208-209, and line 231. Where did you place the LVDT in the experiment to detect the deflection of the deep beam? Please add a test settings image.

Lines 220-221. You wrote, “It concludes that the chosen FE model and inputs are capable of providing reliable results”. Has there been an analysis? What are the parameters? Is there a reference?

Figure 10, check whether the unit of load (Kn) is correct.? Should be kN

Lines 416-418. Look again at the suggestion of using deep beams for infrastructures such as pile caps and shear walls, you must explain how the load in the field and the shear forces occur. Because your research is not completely shear test. If the shear test, the deep beam load should be applied to the corner point resembling an earthquake shear force with a resultant diagonal force.

Line 429, At the "Conclusion", Please give advice on the use of an effective hollow deep beam in the field according to the results of your research.

Author Response

The authors wish to thank the reviewer for the comments and for the timely and thoughtful review. Each comment is addressed individually below.

This topic is quite good, discussing "the behavior of simply supported concrete deep beams reinforced with carbon fiber-reinforced polymer (CFRP)". However, the publication still needs some improvements, including:

Line 19-20, Introduction to the abstract needs consideration for improvement, is it true that roadway culverts, and foundation pile caps have the same behavior as deep beams? If we look at the field the position and loading is very different.

Response: Agree. The authors changed the Phrase (roadway culverts) is changed to shear wall

Line 36, In the abstract, please give advice on the effective use of hollow deep beams in the field according to the results of your research

Response. Agree. The authors elaborated on explaining that the openings are frequently present in deep beams to accommodate doors, windows, various conduits, and communications.

Line 188, what is “Figure 0”, please check.

Response. Agree. The authors removed the phrase “ (Figure 0)”

Line 208-209, and line 231. Where did you place the LVDT in the experiment to detect the deflection of the deep beam? Please add a test settings image.

Response. Agree. The authors added the location of the LVDT in the Figure 2 a.

Lines 220-221. You wrote, “It concludes that the chosen FE model and inputs are capable of providing reliable results”. Has there been an analysis? What are the parameters? Is there a reference?

Response. Agree. The authors elaborated on the explanation that “The comparison between the FE model and the experimental results for three specimens showed a good agreement in both the type of failure and the relation of the load with deflection in figure 4 and figure 5 respectively and it is mentioned in the section 3.3 (Validation of the FE models)”.

Figure 10, check whether the unit of load (Kn) is correct.? Should be kN

Response. Agree. The authors modified the figure as shown below:

Lines 416-418. Look again at the suggestion of using deep beams for infrastructures such as pile caps and shear walls, you must explain how the load in the field and the shear forces occur. Because your research is not completely shear test. If the shear test, the deep beam load should be applied to the corner point resembling an earthquake shear force with a resultant diagonal force”.

Response. Agree. The authors elaborated on explaining that the openings are frequently present in deep beams to accommodate doors, windows, various conduits, and communications.

Line 429, At the "Conclusion", Please give advice on the use of an effective hollow deep beam in the field according to the results of your research.

Response. Agree. The authors elaborated on explaining that the openings are frequently present in deep beams to accommodate doors, windows, various conduits, and communications as follows:

“RC deep beam with opening, reinforced with CFRP stirrups, has many applica-tions such as bridges and buildings. Increasing the size of the opening can alter the stress distribution in the RC deep beam; therefore, it is essential to understand the stress redistribution and tackle the issue by using the CFRP stirrups. When RC deep beam with opening has a failure due to increasing the applied load, CFRP is viable practical solution because it is easy to be installed and maintained.”

Reviewer 3 Report

The paper addresses an important topic that has been researched and documented widely in the literature.

Editorial:

English errors and mistakes and missing data throughout the paper are TOO MANY to list. Given below is just a few examples:

1- The Abstract has English errors/mistakes in almost every sentence; it is needs to be carefully re-written. In addition, the abstract shows 2 numerical values in 2 different systems of units: 2.0 in (line 28) and 160 mm (Line 34). Moreover, it should be CFRP sheets and not CFRP stirrups (line 34).

2- Test specimens in Figures 2 and 7 do not show all dimensions (i.e. location of holes? How far are the holes from the sides, top, and bottom?)

3- Figure 5 has 3 charts. Each chart has 2 curves, but 1 legend is shown. What is the other curve?

The paper should thoroughly be re-written.

Technical:

1- The paper lacks novelty, misses crucial information on test specimens and vital numerical model information, requires more attention to properly report results, and needs extensive English revision.

2- Objective C in Section 2 states: “Investigate the effect of opening size for doubly/singly reinforced concrete deep beam on STM stress behavior.” What is STM stress behavior? The paper seems to miss the effect of opening size on what the authors said “STM stress behavior.”

3- Why did the authors not show STM models for the test specimens with holes? This is important to address since item C above is one of the listed objectives, which seems to be missing.

4- For the parametric study, what are the parameters considered?

5- Figure 9 is meaningless because the legends cannot be read. The reviewer cannot tell which strain is positive and which strain is negative. Therefore, the specimen behavior cannot be explained.

6- Unless a clear, novel contribution is highlighted and the paper is thoroughly re-written and addresses the technical comments above, the paper in its current form does not seem fit for publications.

Author Response

The authors wish to thank the reviewer for the comments and for the timely and thoughtful review. Each comment is addressed individually in the attachment.

Round 2

Reviewer 3 Report

Comments and Suggestions for Authors

The paper addresses an important topic that has been researched and documented widely in the literature.

Editorial:

English errors and mistakes and missing data throughout the paper are TOO MANY to list. Given below is just a few examples:

1- The Abstract has English errors/mistakes in almost every sentence; it is needs to be carefully re-written. In addition, the abstract shows 2 numerical values in 2 different systems of units: 2.0 in (line 28) and 160 mm (Line 34). Moreover, it should be CFRP sheets and not CFRP stirrups (line 34).

Response. Agree. The number 2.0 in (line 28) represent the ratio (b/h) and it is unitless, CFRP Stirrups is replaced with CFRP sheets

Reviewer's comment of the authors response: the number 2.0 in reads as 2 inches. If this is a ration, then it should 2.0 (with no in).

2- Test specimens in Figures 2 and 7 do not show all dimensions (i.e. location of holes? How far are the holes from the sides, top, and bottom?)

Response. Agree. The authors changed the figures to tackle all the mentioned issues as show below.

Reviewer's comment of the authors response: How high, on the sides, are the bars?

3- Figure 5 has 3 charts. Each chart has 2 curves, but 1 legend is shown. What is the other curve?

Response. Agree. The authors changed the figures with figures with two legends as shown bellow:

The paper should thoroughly be re-written.

Response. Agree. The authors conducted a rigor spell, grammatical, punctuation check.

Technical:

1- The paper lacks novelty, misses crucial information on test specimens and vital numerical model information, requires more attention to properly report results, and needs extensive English revision.

Response. Agree. Thank you for your feedback. The present research builds on the existing stream of research related to the development and testing of RC deep beam structure. Based on our intense review of the literature, we believe that this study is unique and the only one of its kind that assesses the RC deep beam behavior based on the experimental, thermotical, and numerical data. The authors also corresponded the change in the behavior to the Strut and Tie Model (STM).

The authors conducted a rigor spell, grammatical, punctuation check.

Reviewer's comment of the authors response: Where is the thermotical data? How does the authors’ model include the temperature? How is this study novel? Where are the details of the numerical model? Where are the STM with the openings?

2- Objective C in Section 2 states: “Investigate the effect of opening size for doubly/singly reinforced concrete deep beam on STM stress behavior.” What is STM stress behavior? The paper seems to miss the effect of opening size on what the authors said “STM stress behavior.”

Response. Agree. The authors elaborated on what (STM) means, strut-and-tie modeling, and it is mentioned in the keywords and it is explained in lines (90-98).

Reviewer's comment of the authors response: Lines 90-98 are typical definitions of STM that can be found in many textbooks. What is STM stress behavior? Why did the authors not show STM models with openings?

3- Why did the authors not show STM models for the test specimens with holes? This is important to address since item C above is one of the listed objectives, which seems to be missing.

Response. Agree. The authors showed how STM is going to be look like in the introduction and elaborated on the meaning and definition as well.

Reviewer's comment of the authors response: The authors response is not clear. Why did the authors not show the analyses of STM models with openings?

4- For the parametric study, what are the parameters considered?

Response. Agree. The authors elaborated on the considered parameters, which were listed in section 2

5- Figure 9 is meaningless because the legends cannot be read. The reviewer cannot tell which strain is positive and which strain is negative. Therefore, the specimen behavior cannot be explained.

Response. Agree. The authors used better Figures with higher resolutions

Reviewer's comment of the authors response: The same problem persists. The legends of figures 9, 11, 12, and 15 are not readable at all. Therefore, the specimen behavior cannot be explained.

6- Unless a clear, novel contribution is highlighted and the paper is thoroughly re-written and addresses the technical comments above, the paper in its current form does not seem fit for publications.

Response. Agree. Thank you for your feedback. The present research builds on the existing stream of research related to the development and testing of RC deep beam structure. Based on our intense review of the literature, we believe that this study is unique and the only one of its kind that assesses the RC deep beam behavior based on the experimental, thermotical, and numerical data. The authors also corresponded the change in the behavior to the Strut and Tie Model (STM). The objectives and novelty of the paper is summarized as follows:

“The main prepossesses of the paper can be listed as follows:

A. Investigate different CFRP strengthening solutions for singly/ doubly RC deep beams on the STM stress web opening.

B. Analyze the effectiveness of the suggested strengthening solutions to restore the beams’ shear resistance through a comparison between laboratory testing and the finite element models, then verify numerical predictions.

C. Investigate the effect of opening size for doubly/singly reinforced concrete deep beam on STM stress behavior and CFRP's effect on the capacity of the structure and stress-strain distributions.”

Reviewer's comment of the authors response: Where is the thermotical data? How does the authors’ model include the temperature? How is this study novel? Where are the details of the numerical model? Where are the STM with the openings and thermal effects?

Author Response

Please find attached file a detailed response for each comment. 

Round 3

Reviewer 3 Report

The reviewer's comments were responded to satisfactorily, except Figure 9b. The STM in Figure 9b is FUNDAMENTALLY wrong: the struts cannot be cut by the cut-off. The paper should not be published with a wrong STM. Please revise Figure 9B with a correct STM and the paper will then be suitable for publication.

Author Response

Please find appended response 
